# Perioperative Management in Neuromuscular Diseases: A Narrative Review

**DOI:** 10.3390/jcm13102963

**Published:** 2024-05-17

**Authors:** Aparna Bhat, Jason Dean, Loutfi S. Aboussouan

**Affiliations:** Respiratory and Neurological Institutes, Cleveland Clinic, Cleveland, OH 44195, USA; bhata2@ccf.org (A.B.); deanj11@ccf.org (J.D.)

**Keywords:** neuromuscular diseases, perioperative care, respiratory function tests, noninvasive ventilation, cardiomyopathies, long QT syndrome, anesthetics, malignant hyperthermia, thymectomy, airway extubation

## Abstract

Patients with neuromuscular diseases are particularly vulnerable in the perioperative period to the development of pulmonary and cardiac complications, or medication side effects. These risks could include hypoventilation, aspiration pneumonia, exacerbation of underlying cardiomyopathy, arrhythmias, adrenal insufficiency, prolonged neuromuscular blockade, issues related to thermoregulation, rhabdomyolysis, malignant hyperthermia, or prolonged mechanical ventilation. Interventions at each of the perioperative stages can be implemented to mitigate these risks. A careful pre-operative evaluation may help identify risk factors so that appropriate interventions are initiated, including cardiology consultation, pulmonary function tests, initiation of noninvasive ventilation, or implementation of preventive measures. Important intraoperative issues include positioning, airway and anesthetic management, and adequate ventilation. The postoperative period may require correction of electrolyte abnormalities, control of secretions with medications, manual or mechanical cough assistance, avoiding the risk of reintubation, judicious pain control, and appropriate medication management. The aim of this review is to increase awareness of the particular surgical challenges in this vulnerable population, and guide the clinician on the various evaluations and interventions that may result in a favorable surgical outcome.

## 1. Introduction

Patients with neuromuscular disease (NMD) may undergo surgery either for general surgical indications or as part of the management of their disease. The latter could include scoliosis or other orthopedic surgeries [1,2], thymectomy for myasthenia gravis (MG) [3], placement of cardio defibrillators or pacemakers in cardiac complication of dystrophies or Friedreich’s ataxia [4], and gastrostomies or tracheostomies in more advanced stages of the disease [5].

Postoperative respiratory complications are common in patients with NMD. The use of opioids, sedative hypnotics, and inhaled anesthetics may aggravate underlying hypoventilation or precipitate it in neuromuscular patients with marginal function (Figure 1) [6]. Bulbar or pharyngeal dysfunction in MG, Guillain–Barré syndrome (GBS), spinal muscular atrophy (SMA), muscular dystrophy, myopathies, and amyotrophic lateral sclerosis, may increase the risk of aspiration. In children with NMD, adverse postoperative events occur in 41% and consist of desaturations consequential enough to require intervention in 73%, airway obstruction in 33%, and atelectasis in 22% [7]. Compared to patients without myasthenia, patients with MG are at a higher risk of postoperative complications including pneumonia, sepsis, bleeding, an intensive care unit stay, prolonged hospitalization, and increased medical expenditure [8].

Surgery may also aggravate the underlying condition. For instance, surgery-related muscle loss has been reported after several types of surgery [9,10]. There is a risk of accelerated progression of disease in amyotrophic lateral sclerosis in the 3 months following surgery [11]. About 35% of myasthenic crises occur following thymectomy [12], leading to respiratory failure requiring mechanical invasive or noninvasive ventilation [13]. 

This review will address the evaluation and management of patients with NMD in the preoperative, intraoperative, and postoperative periods (Table 1).

## 2. Review Strategy

The search was conducted using Medical Subject Heading (MeSH) terms on the PubMed platform including “neuromuscular diseases”, “perioperative care”, “respiratory function tests”, “noninvasive ventilation”, “cardiomyopathies”, “long QT syndrome”, “anesthetics”, “malignant hyperthermia”, “thymectomy”, and “airway extubation”. We combined search terms for relevance to the topic using Boolean operators. Only articles written in the English language were included. Although studies in the past 10 years were favored, there was no restriction regarding the year of publication. Relevant studies, international guidelines, society recommendations, consensus reports, practice guidelines, and expert panel reports published through March 2024 were included.

## 3. Preoperative Assessment

### 3.1. Clinical Presentation

A detailed clinical history is imperative in the preoperative evaluation of a neuromuscular patient. It is not only important to consider the diagnosis but also the duration and the reversibility, symptom stability or severity, and involvement of other organ systems. Clinical considerations in both adults and children include muscle weakness, reduced joint mobility, spinal deformities, respiratory failure, and cardiac and neurologic impairment. Respiratory impairment may involve both inspiratory and expiratory muscles and result in spinal deformities; chest wall contractures, which symptomatically could present as dyspnea; tachypnea at rest; or orthopnea. Certain neuromuscular diseases can be associated with bulbar involvement, which clinically can present as muffled speech, impaired secretions management, aspiration risk, impaired cough, poor coordination of vocal cords, and dysphagia. Cardiac arrhythmias can have high prevalence in myotonic dystrophies. Thus, it is important to elicit a history of fainting, blackouts, and palpitations. After the patient is carefully evaluated with appropriate testing, it is recommended that anesthesiologists, surgeons, pulmonologists, neurologists, and cardiologists work together to develop a safe perioperative and postoperative care plan [14].

Discussions with the patient should include the risk of these complications and the potential need for ventilation invasively or noninvasively in the postoperative period. Should surgery be performed, for example, when it can exacerbate the disease, as can be case with amyotrophic lateral sclerosis where there may be accelerated progression of the disease in the 3 months after surgery [11]?

### 3.2. Physical Exam

The physical exam is a key aspect of preventing postoperative failure in NMD. These patients may present with functional deficits that can complicate their intraoperative management or increase their postoperative risks. Simply observing the patient can identify alterations in functional status, use of accessory muscles for inspiration, or number of breaks they take during conversation. Auscultation can reveal reduced excursion of the diaphragm, crackles because of atelectasis or aspiration, or asymmetric air entry. The presence of these findings may help to identify high-risk patients in a population that is already in a high-risk category for failure to wean from the ventilator [15]. The exam can also assess bulbar muscle weakness with a review of the strength of a voluntary cough, pooling of loose or thick oral secretions, atrophy or fasciculation of the tongue, or a reduced gag reflex [5]. 

### 3.3. Pulmonary Function Testing

The American College of Physicians recommends obtaining forced vital capacity (FVC), maximal inspiratory pressure (MIP), maximal expiratory pressure (MEP), peak cough flow (PCF), SpO_2_, and etCO_2_ for patients with Duchenne Muscular Dystrophy undergoing anesthesia or sedation [16]. These tests may be appropriate in other forms of NMD [17], and an expert panel report also recommends pulmonary function tests to guide treatment decisions in NMD [18]. Sniff nasal inspiratory pressure may be substituted for MIP in those who cannot perform the maneuver [18].

Aspects of the surgical procedure that may place patients with NMD at a disadvantage from a pulmonary perspective include positioning, anesthesia, and surgical incision site. Preoperative pulmonary function tests can help to identify the point where postoperative physiological changes intersect with functional deficits due to an underlying NMD [17]. 

More specifically, pulmonary function tests can assess the magnitude of restrictive impairment, the extent of diaphragm involvement, hypoventilation, and integrity of cough clearing mechanisms.

Assessing the level of supine drop in the vital capacity can determine the extent of diaphragm dysfunction in NMD [17]. A drop of 15–25% in vital capacity or greater when placed in the supine position is seen in unilateral diaphragm impairment (with greater drops in right relative to left diaphragm impairment due to the mass effect of the liver), and >40% in bilateral disorders [17]. A MIP < 60 cmH_2_O is a surrogate marker of nocturnal desaturation, as it is 86% sensitive for the presence of nocturnal O2 saturation nadir of ≤80% [19]. Once these risks have been identified, noninvasive ventilation can be initiated.

Thresholds that may identify an ineffective cough include PCF < 270 L/mn [20,21], or a MEP < 60 cmH_2_O [22]. A critical level for PCF of 160 L/mn, below which clearing the airway may be compromised, has been proposed [20,23]. An expert panel recommends management options including combinations of lung volume recruitment, manually or mechanically assisted cough, and high-frequency chest wall oscillation [18].

### 3.4. Polysomnography

Practice parameters from the American Academy of Sleep Medicine consider polysomnography to be routinely indicated in the evaluation of patients with NMD and sleep-related symptoms that are not otherwise explained [24]. Polysomnography can help distinguish between obstructive, pseudo central, and central sleep apnea events, and identify hypoventilation [25]. 

However, whether preoperative polysomnography screening identifies patients at higher risk, and whether perioperative treatment of sleep-disordered breathing improves outcomes is not well established and recommended for research in guidelines [26]. In a retrospective study that assessed polysomnographic predictors of postoperative complications in children with NMD, only the saturation nadir and the presence of bulbar dysfunction predicted postoperative complications, suggesting that nocturnal oximetry alone may be sufficient as a preoperative evaluation rather than a full polysomnogram [7]. 

Nevertheless, 42% of patients with NMD have sleep-disordered breathing [27]. Given this high prevalence, it is important to screen neuromuscular patients in the preoperative setting. In a study comparing patients with respiratory muscle weakness to subjects without it and patients with obstructive sleep apnea, a score of five or more points on the Sleep-Disordered Breathing in Neuromuscular Disease Questionnaire (SiNQ-5) (score range 0–10 points) had a positive predictive value of 69.4%, and negative predictive value of 95.5% to identify concurrent sleep-disordered breathing and NMD [28]. Comorbidities, such as heart failure, may alter the diagnostic accuracy of the questionnaire [28].

### 3.5. Cardiac Testing

Patients with NMD may have associated cardiomyopathy, arrhythmias, and conduction defects. Patients with both muscular and myotonic dystrophy have a high prevalence of cardiac involvement necessitating assessment with electrocardiogram and echocardiography. Cardiac conduction abnormalities with high prevalence in myotonic dystrophies are atrial fibrillation, atrial flutter, and ventricular fibrillation [29]. If arrhythmias are suspected, then Holter monitoring can be requested.

Patients with ALS [30] and myotonic dystrophy [31] can have a prolonged QT interval. Alternatively, QT interval prolongation is rare in Duchenne dystrophy [32]. Medications used in the treatment of amyotrophic lateral sclerosis such as dextromethorphan with quinidine (Nuedexta) as well as other medications that can be used in patients with NMD (e.g., antidepressants, tricyclics, selective serotonin reuptake inhibitors) can prolong the QT interval and increase the risk of sudden cardiac death from torsade de pointes or ventricular fibrillation.

A scientific statement from the American Health Association recommends the following: a cardiac evaluation before anesthesia or sedation in patients with NMD at risk of cardiac involvement, that the evaluation be within 3–6 months of the anesthesia or sedation event in those with symptoms indicating cardiac involvement, that major surgeries be performed with cardiac monitoring by an anesthesiologist with expertise in NMD, and that these procedures be undertaken in a center with appropriate intensive care facilities [4].

### 3.6. Initiate or Optimize Non-Invasive Ventilation

One important preoperative evaluation is to address whether non-invasive positive pressure ventilation (NIPPV) is provided when appropriate to mitigate postoperative respiratory failure. For instance, less than 53% of patients with ALS were on NIPPV when indicated [33]. An expert panel report recommends initiation of noninvasive ventilation for any of the following: a FVC < 80% with symptoms, FVC < 50% without symptoms, SNIP/MIP < 40 cmH_2_O, or presence of hypercapnia [18]. Reimbursement criteria for initiation of NIPPV in progressive NMD include the following: a PaCO_2_ ≥ 45 mmHg on an arterial blood gas drawn while awake at the prescribed FiO_2_, or an SpO_2_ ≤ 88% for ≥ 5 min of nocturnal recording time (minimum recording time of 2 h) on sleep oximetry undertaken at the prescribed FiO_2_, or either of a MIP < 60 cmH_2_O or FVC < 50% predicted [34]. Once non-invasive ventilation is initiated, settings may be optimized using device downloads and assessment of the PaCO_2_ by arterial blood gases or surrogates (venous, transcutaneous, or end-tidal CO_2_, or bicarbonate level) [35]. We aim to adjust settings such that the pressure support is enough to maintain a respiratory rate of ≤15 [36], and a tidal volume of 8 mL/kg ideal body weight. Nocturnal support of ventilation can maintain better control of daytime CO_2_ [37], and reduce large increases in CO_2_ levels from perioperative decrements in alveolar ventilation from sedation, anesthesia, positioning, atelectasis, pain medication, etc. (Figure) [38].

### 3.7. Preventive Measures

In one review, 41% of patients with NMD were exposed to steroids at one point in their management in conditions such as MG, chronic inflammatory demyelinating polyneuropathy, dermatomyositis, polymyositis, demyelinating polyneuropathy, mononeuritis multiplex, and muscular dystrophy [39]. Stress dose steroids in the perioperative period may be appropriate depending on the dose, duration of use of systemic steroids, and the likelihood of suppression of the hypothalamic–pituitary–adrenal (HPA) axis. When there is uncertainty, a cortisol test or ACTH stimulation test may help assess the integrity of the HPA axis [40]. Stress dose steroids are given immediately prior to surgery, prior to induction or extubation, and continued with a taper postoperatively. Specific protocols are available in the case of Duchenne or Becker muscular dystrophy [40].

In patients with myasthenia, high-dose steroids pre-thymectomy reduced the risk of myasthenic crises in a retrospective study [41] and increased the likelihood of long-term postoperative pharmacologic remission [42]. Patients vulnerable to the development of a myasthenic crisis—such as those with bulbar dysfunction and dysphagia, reduced lung function, or prior history of myasthenic crisis [43,44]—may be candidates for pre-thymectomy prophylaxis with either immunoglobulins or plasma exchange [44].

Malignant hyperthermia (MH) is a hypermetabolic syndrome of severe muscle rigidity (masseter contractures), fever, acidosis, disseminated intravascular coagulopathy, and cardiac arrhythmias. Several hereditary myopathies can increase the risk of MH, including channelopathies such as central core disease, periodic paralysis, and mitochondrial myopathies [45]. Though evidence to support this risk is based on case reports, preoperative dantrolene can be considered in these patients [45].

## 4. Intraoperative Considerations

### 4.1. Positioning

Surgical positioning is key in hereditary neuropathy with liability to pressure palsy (HNPP) to avoid entrapment neuropathies [46]. The most common sites are the fibular head and cubital tunnel [47], and prolonged foot drop has been reported as a complication of total knee arthroplasty in HNPP [48].

In NMD associated with autonomic nerve dysfunction, there can be significant variability in the vital signs depending on surgical positioning [45]. Autonomic dysfunction can be seen in MG [49], muscular dystrophies [50], and amyotrophic lateral sclerosis [51].

### 4.2. Anesthetic Considerations

#### 4.2.1. Airway

Depolarizing neuromuscular blocking agents can potentiate myotonia and complicate bag mask ventilation and tracheal intubation. Macroglossia, commonly seen in Pompe disease [52], and less frequently in muscular dystrophies [53], may be an important challenge in rapid sequence intubation. Awake intubation or use of a hyper-angulated blade can mitigate the challenges of intubation. Control of salivary, pharyngeal, or tracheobronchial secretions with pre- or intraoperative glycopyrrolate in patients with NMD may help facilitate intubation, though contraindicated in patients with MG [54].

#### 4.2.2. Anesthesia and Cardiac Function

A scientific statement from the American Health Association recommends (Level 1, Class C evidence) that major surgeries in patients with NMD be performed with cardiac monitoring by an anesthesiologist with expertise in NMD, and that these procedures be undertaken in a center with appropriate intensive care facilities [4]. In NMD patients with cardiomyopathy, there should be careful monitoring given the potential myocardial depressing properties of inhalational and intravenous anesthetics, and the risk of prolongation of the QTc interval. Specifically, several agents may prolong the QTc (propofol, etomidate, ketamine, thiopental, and volatile anesthetics such as sevoflurane, isoflurane, desflurane). Opioids may be safer in this regard than propofol, but require careful attention to the risk of hypoventilation [55].

#### 4.2.3. Choice of Anesthetics

The risks of general anesthesia in NMD include rhabdomyolysis, prolonged myotonia, hemodynamic instability, malignant hyperthermia, and prolonged postoperative mechanical ventilation [56].

Neuromuscular blocking agents are commonly used for endotracheal intubation, mechanical ventilation, and to facilitate surgical interventions. There are two types of neuromuscular blocking agents based on their mechanism of action, depolarizing and non-depolarizing blocking agents.

Succinylcholine, the only available depolarizing neuromuscular blocking agent, should be avoided in neuromuscular disorders due to the risk of exacerbating pre-existing muscle weakness and uncontrolled muscle contractions after depolarization. Hyperkalemia due to the efflux of potassium from muscle can lead to life-threatening arrythmias and sudden cardiac death. The use of succinylcholine in peripheral nerve disorders, such as Guillain–Barré syndrome and motor neuron disease is also not recommended due to the upregulation of acetylcholine receptors that is observed which can in turn intensify potassium efflux [14,57]. Unique to MG is the destruction/downregulation of acetylcholine receptors which leads to a resistance to succinylcholine and increased dosage requirements of up to 2.6 times [3]. It is also important to note the possibility of interactions with azathioprine which can prolong the effects of succinylcholine and inhibit non-depolarizing agents [58,59].

Non-depolarizing neuromuscular blocking agents are considered safer in NMD, though there is increased sensitivity in NMD patients of various etiologies [60,61]. Since these patients have reduced muscle force and mass, they may require a 10–20% lower dose of neuromuscular blockade and experience a prolonged duration of action [62]. Thus, it is important to monitor the effects of neuromuscular blockade with intra- and perioperative neuromuscular monitoring or train-of-four monitoring to evaluate the duration of the effect [14].

If there is residual muscle relaxation postoperatively, then either a selective muscle relaxant antagonist or a cholinesterase inhibitor may be used. Sugammadex is the most used selective muscle relaxant antagonist with a rapid onset and complete reversal of nondepolarizing neuromuscular blocking agents, which may prevent the need for prolonged postoperative ventilation [57,63], though this benefit is not always assured [64]. Cholinesterase inhibitors may also be used with anticholinergics, though this is not recommended in muscular dystrophy due to hyperkalemia and cholinesterase inhibitor-induced myotonia. If neither option is available, then the patient can remain intubated and mechanically ventilated until the effects of neuromuscular blocking agents resolve.

Anesthesia-induced rhabdomyolysis with volatile anesthetics can be seen in muscular dystrophies causing hyperkalemia and sudden cardiac arrest [14]. The risk of rhabdomyolysis with administration of volatile anesthetics in myopathies could be related to the disruption of the muscle cell membranes [62]. Finally, etomidate should be avoided in patients at risk of adrenal suppression and adrenal crisis [65].

#### 4.2.4. Alternative Approaches

Alternatives to general anesthesia include local or regional anesthesia (epidural, spinal or regional nerve block) (Table 2) with the use of sedation medications, such as midazolam, fentanyl, ketamine, and propofol [62]. Regional anesthesia can be used in patients with junctional and post-junctional neuromuscular disease since there is still intact neuronal function. However, patients with pre-junctional disorders have neuronal disruption and thus would not benefit from regional anesthesia. It should be noted that benzodiazepines and opioids can both result in respiratory depression, airway obstruction, and worsening of muscle weakness. Thus, their use should be limited to lower doses and only used with judicious monitoring of pulse oximetry, capnography, and respiratory rate [14].

#### 4.2.5. Thermoregulation

Anesthetics can lower the core body temperature intraoperatively, particularly in patients with reduced muscle mass, due to impairment in thermoregulatory responses resulting in reduced heat generation, increased heat dissipation, and alteration in other compensatory mechanisms [66]. There may be impairments in evoked electromyographical response of the diaphragm with hypothermic cardioplegia to about 31 °C during cardiopulmonary bypass [67]. Hypothermia can further exacerbate myotonia and the sensitivity to sedatives and non-depolarizing neuromuscular blockade [68]. For instance, twitch responses of the adductor pollicis decrease about 10% for each °C drop and up to 20% for each ºC drop in the presence of vecuronium [68]. Hypothermia can also reduce the elimination rate of muscle relaxants [68]. Thus, it is important to closely monitor the core body temperature perioperatively [14].

Malignant hyperthermia (MH), mentioned earlier, can be precipitated due to a calcium imbalance in skeletal muscle with the use of volatile anesthetics (i.e., halothane, sevoflurane, isoflurane, and desflurane), succinylcholine, or the combination of both [14]. The anesthesiologist should maintain a high index of suspicion to administer dantrolene as soon as the symptoms are recognized [45,60,69].

#### 4.2.6. Effects of Anesthesia on Pulmonary Function and Ventilator Management

Even in a patient without diaphragm paralysis, anesthesia results in a cephalad displacement of the dependent portion of the diaphragm [70], with stable position or caudal shift of the non-dependent area [71]. Furthermore, there is a reduction in FRC that is not entirely explained by the actual displacement of the diaphragm. For instance, pancuronium decreases the FRC in proportion to inspiratory muscle weakness [72]. The decrease in the FRC is noted shortly after anesthesia and is present with both inhalational or intravenous anesthetics as well as with spontaneous or mechanical ventilation [73].

There are also changes in lung distensibility and lung and chest wall elastic properties that decrease total respiratory system compliance (Crs) [74] and are postulated to be a result of rapid and shallow breathing pattern with increasing work of breathing that results in chronic micro atelectasis and decrease in both lung and chest wall compliance [5]. Reduced Crs may necessitate higher pressures to achieve adequate ventilation and tidal volumes with mechanical ventilation. End-tidal CO_2_ monitoring or waveform capnography can be a useful method to monitor for adequate ventilation intraoperatively. In a systematic review of non-cardiac surgery in a general population, lung-protective intraoperative ventilation conferred benefits with reduction in postoperative complications, though the quality of the evidence was of low or moderate quality [75].

## 5. Postoperative Care

### 5.1. General Care

Interventions to mitigate the potential adverse effects of hypoxemia, hypoventilation, and difficulty with secretion clearance include deep breathing exercises (diaphragm exercises, pursed-lip breathing, postural changes to favor ventilation, upper and lower limb exercises combined with breathing), postural drainage, percussion, lung expansion interventions (such as incentive spirometry and intermittent positive-pressure breathing), and PAP therapy [75,76,77].

Postoperative positional changes that may be of particular importance in NMD include upright or semi recumbent positioning in those with orthopnea and supine drop in vital capacity, and preferentially adopting a dependent position for the most affected diaphragm in asymmetric diaphragm involvement [25]. Note that there may be significant exceptions to these general guidelines: C4 to C7 tetraplegia can be associated with platypnea and orthodeoxia with an increase in supine vital capacity [78]. Additionally, in contrast to asymmetric diaphragm impairment, in the setting of a unilateral infiltrate or atelectasis, improvement in ventilation perfusion patching and oxygenation can occur with the healthier lung in a dependent position [79].

Intraoperative blood loss and fluid replacement therapy may result in electrolyte imbalances. Metabolic factors that may impair neuromuscular function such as hypothermia [80], acidosis [81], hypophosphatemia particularly in older individuals [82], hypomagnesemia, hypocalcemia, and hypokalemia should be monitored and corrected.

Glycopyrrolate can impair heat dissipation and increase body temperature, particularly if there is an underlying autonomic neuropathy, postoperative fever, increased ambient temperature, or after exercise [83].

### 5.2. Management of Secretions

Secretion management is paramount in neuromuscular patients given the varying ability to self-manage secretions in this population. About 20–40% of patients with motor neuron disease have sialorrhea [84]. This may be compounded by dysphagia in up to 60% of patients (even in the absence of NMD) following prolonged intubation (up to 5 days) [85].

One first step to secretion management is determining whether secretions are thick or thin. Thin secretions can be managed with oral suctioning with a suction pump and catheter to remove oral secretions. In some cases, anticholinergics can reduce the quantity of saliva. These medications need to be used carefully, especially in patients with myasthenia, where they can aggravate the NMD, or in Parkinson’s due to the risks of mental side effects including confusion, restlessness, or hallucinations [86].

Thick secretions can be thinned with mucolytics, such as guaifenesin or N-acetylcysteine. In a systematic review, prophylactic mucolytics had a favorable effect on postoperative pulmonary complications in a general population [75]. Although the quality of the evidence was considered of low or moderate quality, the impact of mucolytics may be more marked in patients with NMD. Mechanical interventions for thick secretions may include oscillating vest therapy or a cough-assist device [5].

### 5.3. Weaning, Extubation and Positive Airway Pressure Therapy

A systematic review noted that there were too few studies to make recommendations about specific weaning protocols in NMD [87]. However, patients with NMD are at increased risk of extubation failure from postoperative respiratory failure and aspiration, particularly in those with ineffective cough with peak cough flows < 160 L/min [20,23], hypercapnia with spontaneous breathing trials [88], and prior extubation failure [89].

Noninvasive ventilation may mitigate this risk, especially when combined with measures to facilitate secretion clearance. In a randomized trial of patients with chronic respiratory disorders and hypercapnia during a spontaneous breathing trial after extubation, early/prophylactic use of noninvasive ventilation reduced subsequent respiratory failure and 90-day mortality [88]. A randomized trial of this sort would be difficult to implement in patients with NMD [89], but noninvasive ventilation and manually or mechanically assisted coughing reduced the risk of reintubation or tracheostomy compared to historical controls in NMD [15]. This study utilized a control and intervention group both with 10 patients. While the sample size of this study was small, there was a significant reduction in reintubation and tracheostomy placement, 3 vs. 10 and 3 vs. 9, respectively. In addition, those extubated to NIPPV had a much shorter stay in the intensive care unit [15].

### 5.4. High Flow Nasal Cannula

Nasal High Flow (NHF) is a method of oxygen delivery at high flow rates with heated humidification. It may reduce dead space ventilation, as well as a component of CO_2_ rebreathing with improved comfort relative to noninvasive ventilation [90]. In a study of patients at elevated risk of reintubation failure (but which did not include patients with NMD), high-flow therapy was not inferior to noninvasive ventilation, with prevention of postextubation respiratory failure in 26.9% with NHF compared to 39.8% with noninvasive ventilation [91]. In contrast, one study found that overnight NHF in neuromuscular patients with sleep-disordered breathing did not improve the transcutaneous CO_2_ level, apnea hypopnea index, or oxygen saturation levels, and higher flow rates (50 L per minute) were poorly tolerated [90]. In another study, 24 h use of high-flow nasal cannulas in three patients with NMD who were intolerant of NIV resulted in treatment failures including intubation and severe respiratory acidosis [92]. A combination of NIV at night with HFNC during the daytime did seem to be more effective [92].

### 5.5. Pain Control

Control of post-surgical pain may be challenging in patients with NMD as they may be at increased risk of complications with therapeutic dosing of medication. Specific complications include pain from post-surgical spasticity, worsening weakness, respiratory depression, and worsening dysphagia. More specifically, patients with NMD and at risk of hypoventilation may be particularly sensitive to the use of opioids, barbiturates, and benzodiazepines (Figure). Alternative management options may include regional pain control with epidural or peripheral nerve blockade (Table 2), use of short-acting rapid-onset opioids, and substitution with non-steroidal drugs.

### 5.6. Pharmacological Considerations

A review of preoperative medications will help plan for potential complications associated with withdrawal effects in the postoperative period, or address interactions with common postoperative drugs.

Worsening sialorrhea, if oral anticholinergic agents are withheld, can be mitigated by atropine eye drops, or scopolamine patches. Oral mucolytics can be changed to nebulized alternatives such as nebulized hypertonic saline or N-acetylcysteine. Discontinuation of baclofen or antispasmodic may cause withdrawal symptoms including the baclofen-withdrawal syndrome and can have adverse results including progressive weakness and respiratory insufficiency [93]. Potential alternatives include targeted botulinum injection and physiotherapy [94].

Antiemetics for the management of postoperative nausea and vomiting such as ondansetron, droperidol, haloperidol, or metoclopramide can prolong the QT interval, particularly in patients at risk (see cardiac testing section above). For these patients, dexamethasone may be an appropriate substitute [55].

The surgical and medical team involved in the care of patients with myasthenia should be aware that commonly prescribed postoperative medications can precipitate or worsen myasthenic crisis. The list is extensive and includes aminoglycosides, quinolones, macrolides, magnesium, beta blockers, procainamide, quinidine, antipsychotics with anticholinergic effects, antiarrhythmics, and local anesthetics [62,95]. Post-surgical complications including pulmonary infection may further compound the risk of a myasthenic crisis [96]. An international executive summary considers immunoglobulin therapy or plasma exchange to be equally effective in the management of a myasthenic crisis. Although immunoglobulins are easier to administer, plasma exchange is preferred because of a faster onset, and may be more effective than immunoglobulins in MusK positive, milder, or ocular myasthenia. Other considerations that may determine the choice are that patients with sepsis cannot receive plasma exchange, and immunoglobulins cannot be given in renal failure [44].

For patients who had been on steroids preoperatively and who have not received stress dose steroids, a random cortisol may be appropriate in the setting of symptoms of abdominal pain, nausea, vomiting, hypotension, hyponatremia, or hypokalemia, particularly in the presence of additional postoperative stressors including infection, dehydration, bleeding, or surgical complications (such as re-exploration) [40]. More urgent situations may justify empiric treatment with steroids until the etiology is ascertained. Intravenous steroids may be necessary in those unable to take oral medications postoperatively. Note that high-dose steroids can result in critical illness myopathy postoperatively, particularly when a non-depolarizing neuromuscular blocking agent is used with intubation [45].

### 5.7. Postoperative Pulmonary Function Tests

A nadir in vital capacity is noted on the first postoperative day for most procedures, though it could be on the second postoperative day for non-resectional thoracotomy [97]. A return to baseline may be delayed for as long as one week in upper abdominal surgery [97]. General care measures outlined above may be helpful to mitigate the potential complications.

The role of routine postoperative pulmonary function testing is difficult to ascertain. Most of the studies on patients with NMD apply to inpatient management of patients with GBS or MG to assess the risk of respiratory failure. In GBS, respiratory function tests that were associated with a risk of requiring mechanical ventilation included a vital capacity < 20 mL/kg or <60%, a reduction in vital capacity of >30%, MEP < 40 cmH_2_O, or MIP < 30 cmH_2_O [98,99]. In MG, a reduction in MIP by ≥30% was associated with a risk of requiring invasive or noninvasive ventilation, whereas favorable factors include a vital capacity > 20 mL/kg maximal inspiratory or expiratory pressure > 40 cmH_2_O [100]. In another study, repeated measurements of vital capacity did not predict the need for mechanical ventilation in MG [101].

## 6. Conclusions

Patients with NMD are at risk of perioperative respiratory complications (including hypercapnic and/or hypoxemia respiratory failure, aspiration, atelectasis), and cardiac complications (heart failure, arrhythmias, conduction abnormalities). Preoperative evaluation with appropriate assessment; intraoperative consideration including positioning, choice of anesthetics, and ventilator management; and postoperative care including weaning, prevention, pain control and medication considerations can help improve the outcomes.

## Figures and Tables

**Figure 1 jcm-13-02963-f001:**
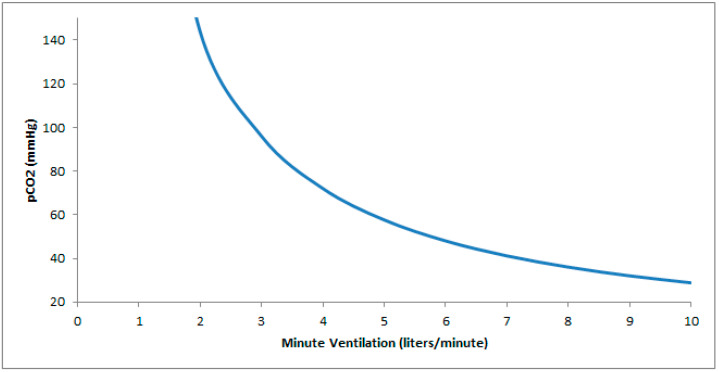
Hyperbolic relationship between pCO_2_ and ventilation. In patients with neuromuscular disease and a baseline low minute ventilation, a minor drop in ventilation (as can occur with atelectasis, sedation, positional changes, or anesthesia) results in a significantly greater increase in pCO_2_ relative to an equivalent drop occurring at a higher minute ventilation.

**Table 1 jcm-13-02963-t001:** Findings and potential actions in the perioperative management of patients with neuromuscular disease.

	Preoperative		Intraoperative		Postoperative	
	Test or Finding	Action	Test or Finding	Action	Test or Finding	Action
Pulmonary Function tests	Sitting VC < 50%, MIP < 60 cmH_2_O, CO_2_ ≥ 45 mmHg or nocturnal SpO_2_ ≤ 88% for ≥5 min	Initiate NIPPV	See ventilation		Reduced VC post-op	Deep breathing, postural drainage, percussion, and lung expansion
					Post-op testing for GBS or MG. Reduction of MIP or VC by >30%	Identify risk of reintubation: step up medical therapy
Cardiac	Dystrophies: Risk of arrhythmias, conduction abnormalities	Cardiac evaluation. Holter test prior to anesthesia or sedation	Dystrophies: Risk of arrhythmias, conduction abnormalities	Cardiac monitoring by anesthesiologist with expertise in neuromuscular diseases	Antiemetics that prolong QT: ondansetron, droperidol, haloperidol, or metoclopramide	Monitor QT, consider dexamethaso-ne instead
	Risk of prolonged QT: medications, ALS, myotonic dystrophy	EKG, cardiac evaluation	Drugs that prolong QT: Propofol, etomidate, ketamine, thiopental			
Positional issues	Orthopnea and supine drop in VC	Sleep upright or with a wedge pillow	HNPP	Position to avoid nerve entrapment	Check radiograph for aspiration, atelectasis pneumonia	If hypoxemic, try to position with healthier side down (improve V/Q)
	Trepopnea	Keep affected diaphragm dependent	Autonomic dysfunction (MG, dystrophy, ALS)	Check vital signs with positioning		
Secretions	Bulbar dysfunction MEP < 60 cmH_2_O, PCF < 270 L/mn	Initiate secretion clearance techniques	Salivary, pharyngeal, or tracheobronchial secretion	Glycopyrrolate to facilitate intubation	High aspiration risk. Increased dysphagia after intubation	Manual or mechanical assisted coughing
Ventilation	On NIPPV: Tidal volume < 8 mL/kg IBW, respiratory rate > 15, CO_2_ > 45, AHI > 5	Increase EPAP or PS	Reduced FRC and respiratory system compliance	Higher ventilation volume/pressure Monitor etCO_2_	High reintubation risk: Hypercapnia on SBT, prior extubation failure, aspiration risk	Extubate to NIPPV, usually combined with assisted coughing
Thermoregulation	Risk for MH in hereditary myopathies	Consider preop dantrolene	Volatile anesthetics may precipitate MH	Dantrolene	Glycopyrrolate may impair heat dissipation	Monitor core body temperature
			Risk of hypothermia	Monitor core temp		
Pharmacologic	On chronic steroids	Consider stress dose steroids	Recognize risk of rhabdomyolysis, prolonged myotonia, MH. Increased sensitivity to non-depolarizing agents.	Avoid succinylcholine. Sugammadex for complete reversal	Signs and symptoms of adrenal insufficiency	Empiric steroid or based on cortisol
	MG at risk of crisis: bulbar dysfunction, reduced lung function, prior crisis	Pre-thymectomy prophylaxis with immunoglobulins/plasma exchange			MG crisis	Immunoglobulin therapy (easier) or plasma exchange (faster)

Abbreviations. AHI: Apnea hypopnea index, FRC: functional residual capacity, IBW: ideal body weight, HNPP: hereditary neuropathy with pressure palsy, GBS: Guillain–Barré syndrome, MEP MIP: maximal expiratory and inspiratory pressure, MG: myasthenia gravis, MH: malignant hyperthermia, NIPPV: noninvasive positive pressure ventilation, VC: Vital capacity, V/Q: ventilation perfusion.

**Table 2 jcm-13-02963-t002:** The use of regional anesthesia in neuromuscular disease intraoperatively and for postoperative pain control based on surgical intervention.

	Intraoperative Anesthesia Postoperative Pain Control	Selected Indications
Gastrointestinal	Epidural, spinal, paravertebral nerve blocks, transversus abdominis plane block	Colon resections, cholecystectomy, stomach, intestinal, liver and hernia repair
Gynecology	Epidural, spinal, paravertebral nerve blocks, transversus abdominis plane block	Hysterectomy, pelvic procedures including prolapse, myomectomy, oophorectomy, Cesarean sections
Obstetrics	Epidural, spinal, paracervical and pudendal blocks	Labor and vaginal deliveries
Ophthalmology	Retrobulbar, peribulbar, or episcleral block	Cataract removal LASIK Trabeculectomy Vitreoretinal surgeries
Orthopedics/limb procedures	Epidural, spinal, and peripheral nerve blocks, intravenous regional anesthesia (Bier block)	Bone and joint surgeries, including replacements Shoulder or arm surgery: Brachial Plexus Block Knee surgery: femoral or sciatic and popliteal Achilles, Ankle, Foot surgery: Sciatic and Popliteal Hand or wrist: Bier block
Thoracic and chest wall	Epidural, thoracic paravertebral, facial plane blocks (e.g., pectoralis, serratus), intercostal nerve blocks	Chest, breast, and esophageal surgeries
Urology	Epidural, spinal, paravertebral nerve blocks, transversus abdominis plane block, pudendal block	Radical prostatectomy, transurethral resection of prostate, nephrectomy, lithotomy, lithotripsy
Vascular	Epidural or paravertebral block, cervical plexus block	Carotid endarterectomy (cervical plexus block) Abdominal aortic endovascular procedure Lower extremity graft bypass

## Data Availability

No new data were created or analyzed in this study. Data sharing is not applicable to this article.

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
