# Peer review of "Perioperative Management in Neuromuscular Diseases: A Narrative Review"

_jcm, 2024, doi:10.3390/jcm13102963_

Round 1
Reviewer 1 Report
Comments and Suggestions for Authors
This manuscript is a very well written review article which attempts to elucidate all aspects of pre-operative and post-operative management of patients suffering from neuromuscular disease. All relevant risk factors are clearly and adequately presented and analyzed, along with all available methods aiming towards the prevention of post-operative complications. An extensive bulk of literature evidence is included and all aspects of pre-operative evaluation and intra-operative issues are addressed in detail. The figure and table that accompanies the manuscript is really helpful and enhances the overall merit and scientific soundness of the article. The conclusion section entails a comprehensive report centerd on the perioperative respiratory complications of patients with neuromuscular disease, along with all necessary considerations in order to help improve the outcomes.
Author Response
Thank you for your comments. They are greatly appreciated
Reviewer 2 Report
Comments and Suggestions for Authors
Dear authors,
Thank you for submitting your work. However, I have a few comments and I expect it to be submitted after necessary edits to process it further and make it more comprehensive.
Please add in the abstract why you intended to draft this write-up— recent updates, bridging the knowledge gap, awareness among clinicians- surgeons and anesthesiologists, etc.
The title should mention the type of article (narrative review, scoping review). Please add it to attract more readers and citations (subject to acceptance).
May I suggest including muscular dystrophies in the second paragraph of the introduction?
In the preoperative assessment of nerve conduction studies, please mention EMG as an aid to diagnose NMD.
In 4.5 (pain control) summarize the regional anesthesia options in a tabular form based on the type of surgery.
After the introduction, the authors jumped to preoperative assessment. I suggest having a heading- Review strategy, in which it is suggested to mention databases searched, keywords used, and duration of search (from this year to date). This will make the manuscript more systematic, and the flow will be maintained.
Resubmit after making relevant changes.
Comments on the Quality of English LanguageThe English language used is fine. If accepted, the article will need technical edits.
Author Response
Thank you for your comments. They reflect a thorough review of the manuscript and propose changes that will enhance the work. In response we have attached an updated manuscript. A point-by-point response is listed below.
Please add in the abstract why you intended to draft this write-up— recent updates, bridging the knowledge gap, awareness among clinicians- surgeons and anesthesiologists, etc.: This was addressed
The title should mention the type of article (narrative review, scoping review). Please add it to attract more readers and citations (subject to acceptance).: Title was revised to include the type of article.
May I suggest including muscular dystrophies in the second paragraph of the introduction?: Thank you for that useful comment. We have revised the sentence referenced in this comment to include conditions associated with pharyngeal dysfunction (rather than from bulbar dysfunction alone): muscular dystrophies and SMA were added.
In the preoperative assessment of nerve conduction studies, please mention EMG as an aid to diagnose NMD.: We did not make this change as the diagnosis of NMD is beyond the topic of our review. We actually wondered whether this comment was intended for our review as we do not discuss nerve conduction studies either. All the interventions we listed (PFT, sleep studies, cardiac testing noninvasive ventilation) have specific relevance to the perioperative evaluation of NMD patient. A literature review does show utility of perioperative electrodiagnostic techniques in radiculopathy, plexopathy, neuropathy, for nerve or spinal surgery, but to the best of our knowledge this was not specific to NMD.
In 4.5 (pain control) summarize the regional anesthesia options in a tabular form based on the type of surgery.: We added a Table 2 and renumbered the original Table as Table 1. That table applies to both intraoperative and postoperative sections
After the introduction, the authors jumped to preoperative assessment. I suggest having a heading- Review strategy, in which it is suggested to mention databases searched, keywords used, and duration of search (from this year to date). This will make the manuscript more systematic, and the flow will be maintained.: Thank you for that comment. A review strategy was included, and the numbering of subsequent sections was accordingly revised
Resubmit after making relevant changes.
Round 2
Reviewer 2 Report
Comments and Suggestions for Authors
Dear authors,
Thank you for revising this manuscript based on the comments raised. I am very happy to see the present version of the manuscript, including the title change.
Best wishes.
Comments on the Quality of English LanguageThe use of English language is fine.